# Contrary to the Conclusions Stated in the Paper, Only Dry Fat-Free Mass Was Different between Groups upon Reanalysis. Comment on: “Intermittent Energy Restriction Attenuates the Loss of Fat-Free Mass in Resistance Trained Individuals. A Randomized Controlled Trial”

**DOI:** 10.3390/jfmk5040085

**Published:** 2020-11-20

**Authors:** Jackson Peos, Andrew W. Brown, Colby J. Vorland, David B. Allison, Amanda Sainsbury

**Affiliations:** 1Faculty of Science, School of Human Sciences, The University of Western Australia, Crawley, WA 6009, Australia; 2Department of Applied Health Science, School of Public Health-Bloomington, Indiana University, Bloomington, IN 47405, USA; awb1@iu.edu (A.W.B.); cvorland@iu.edu (C.J.V.); 3Department of Epidemiology and Biostatistics, School of Public Health-Bloomington, Indiana University, Bloomington, IN 47405, USA; allison@iu.edu

**Keywords:** diet, caloric restriction, refeed, diet break, nutrition, sports nutrition, fat loss, weight loss, lean mass, resting energy expenditure, resting metabolic rate, athlete

Campbell and colleagues recently published a randomised controlled trial investigating the effects of diets involving intermittent energy restriction versus continuous energy restriction on changes in body composition and resting metabolic rate (RMR) in resistance-trained adults [1]. Intermittent energy restriction is a topic of great interest to athletes, and this was a greatly needed study, because previous research in the realm of intermittent energy restriction has focused on populations with overweight or obesity. Moreover, the diet that Campbell and colleagues tested in their trial (7 weeks with 5 days per week of prescribed moderate energy restriction and 2 weekend days per week with no prescribed energy restriction) is of relevance to many athletes who implement weekend ‘refeeds’ or ‘diet breaks’ during energy restriction regimes in an effort to prevent loss of fat-free mass (FFM) and reductions in RMR [2]. In keeping with the hypothesised effects of intermittent energy restriction, the authors concluded that “a 2-day carbohydrate refeed preserves FFM, dry FFM and RMR during energy restriction compared to continuous energy restriction in RT (resistance-trained) individuals.” However, we have two concerns regarding the statistical analysis and interpretation of the data, and these concerns lead us to believe that re-evaluation of the authors’ conclusion is warranted. Our first concern is that the authors have drawn conclusions from their data based on differences in nominal significance (i.e., the “DINS error”, detailed below), rather than on actual differences between the diet groups. Our second concern is that the authors have based their conclusion on analysis of only those people who completed the trial (a completers-only analysis, also known as a per-protocol analysis), rather than also analysing all people who were randomised to the diets (intention-to-treat analysis, also detailed below).

The DINS error in randomised controlled trials occurs when authors wrongly conclude that the randomised group results demonstrate an effect of treatment assignment when one group shows a significant change from baseline, but another group does not. For example, Campbell et al. conclude that the intermittent/refeed diet results in greater retention of FFM and RMR than the continuous diet, presumably because there were statistically significant reductions in FFM and RMR relative to baseline (pre-diet) in the continuous diet group, but no statistically significant pre-diet to post-diet changes in the intermittent/refeed diet group, as reported in their Table 3. In other words, they appear to have interpreted the significant within-group decreases from baseline in FFM and RMR in the continuous diet group (but not in the intermittent/refeed diet group) as evidence of a significant between-group difference. This analytic strategy is invalid, as noted previously in the literature [3,4,5]. The DINS error can increase the chance of false positive results (i.e., concluding that there is a difference between groups when in fact no difference exists) to up to 50% (and even higher if group sample sizes are unequal), compared to the usually-accepted false positive rate of 5% (as is the case when statistical significance is set at *p* < 0.05) [4,5,6]. Multiple papers using this invalid analytic strategy have had to be corrected [7,8,9] or retracted [10,11].

In parallel-group randomised controlled trials, the correct way to determine whether treatment effects are different from each other is to directly compare outcomes in the treatment and control groups [3,12]. Campbell et al. undertook these direct comparisons between the intermittent/refeed diet and the continuous diet groups using a two group × two time point between–within factorial analysis of variance (ANOVA) with repeated measures. Contrary to their conclusion, there was no significant difference between diet groups for the change in FFM or RMR, albeit they did report a significantly greater preservation of dry FFM in the intermittent/refeed diet group. To investigate this further, we analysed the data provided in the publication’s online supplementary material using analysis of covariance (ANCOVA) with baseline values as a covariate, which has been shown to generally provide greater power than ANOVA in randomised controlled trials [13]. Like Campbell et al., we also found no significant difference between the intermittent/refeed and continuous diet groups for the change in FFM or RMR, albeit we did also observe significantly greater retention of dry FFM upon completion of the intermittent/refeed diet versus the continuous diet (*p* = 0.0004).

In addition to the DINS error, the paper by Campbell et al. does not report any intention-to-treat analysis, instead analysing data only from participants who completed the 7-week diets (a completers-only analysis). Considering that 27 of 54 (50% of) participants that underwent baseline testing completed the diet and trial as intended, those that withdrew or were excluded from the trial represent a large proportion of the original sample size. Completers-only analyses have been shown to overestimate the effects of interventions and have led to retractions [14,15], because they only examine those participants who remained in the trial, with these participants being more likely to have adhered to the intervention than those who withdrew from the trial. This is an important point, because non-compliance and withdrawal from interventions is likely to happen in actual practice [16]. To investigate the possibility that the completers-only analyses conducted by Campbell et al. may have overestimated treatment effects, we undertook an intention-to-treat analysis on the three outcomes mentioned above—FFM, dry FFM, and RMR (see our calculations at https://doi.org/10.5281/zenodo.3961834), using the Baseline Observation Carried Forward (BOCF) approach. BOCF is less robust than approaches such as multiple imputation, but was the only approach available to us given lack of data on the 27 of 54 participants that withdrew or were excluded from the trial. We conducted our analysis on changes from baseline, assuming that missing participants had no change from baseline [17]. Specifically, we compared the two diets groups in terms of the change from baseline in each of the three outcomes, using a between-group Welch’s *t*-test. We used a Welch’s *t*-test rather than a Student’s *t*-test, because the Welch’s *t*-test does not assume normality of the data (unlike a Student’s *t*-test), and with half of the data set having a value of zero (i.e., no change from baseline), the data were unlikely to be normally distributed. It has been argued that the Welch’s *t*-test should be used by default rather than Student’s *t*-test [18]. For FFM and RMR, we saw no significant differences between diet groups in change scores (FFM *p* values for Welch’s *t*-test on completers-only and BOCF data of 0.1795 and 0.1656, respectively, and RMR *p* values of 0.4916 and 0.4452, respectively). For dry FFM, we again detected a significant between-group difference, but the magnitude of the difference between diet groups was smaller with the BOCF analysis than with the completers-only analysis. Specifically, for the BOCF analysis, dry FFM was significantly greater in the intermittent/refeed diet group compared to the continuous diet group, by 0.9 kg (95% confidence interval 0.33 to 1.46 kg; *p* = 0.0028), which is about half the estimate from the completers-only analysis, where the difference was 1.7 kg (95% confidence interval 0.86 to 2.57 kg; *p* = 0.0004).

We commend the authors for sharing their data as a supplement in the spirit of more transparent and reproducible science. Their sharing of data enabled us to confirm that the effects of intermittent versus continuous energy restriction on dry FFM in resistance-trained adults seem to be robust to completers-only ANCOVA, completers-only Welch’s *t*-test on change scores, and to a simplistic (BOCF) intention-to-treat analysis, also using Welch’s *t*-test on change scores. Although the latter changed the magnitude of the estimated effect size, the *p* value was less than a Bonferroni-corrected alpha for multiple outcomes, demonstrating statistical significance. Conversely, we were able to determine that the comparison of within-group changes using the DINS approach resulted in erroneous conclusions about the effects of an intermittent/refeed diet versus a continuous diet on FFM and RMR. Thus, the original conclusions of Campbell et al. should be corrected to indicate that only one of these outcomes (dry FFM) was significantly different between the intermittent/refeed and continuous diets. Upon correction, we also request that *p* values be reported exactly, as per statistical reporting guidelines [19] (i.e., provide exact *p* values instead of statements such as *p* < 0.05), which will further aid readers in interpretation of the data.

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
