# Peer review of "Contrary to the Conclusions Stated in the Paper, Only Dry Fat-Free Mass Was Different between Groups upon Reanalysis. Comment on: “Intermittent Energy Restriction Attenuates the Loss of Fat-Free Mass in Resistance Trained Individuals. A Randomized Controlled Trial”"

_jfmk, 2020, doi:10.3390/jfmk5040085_

Round 1

Reviewer 1 Report

This LTE adds reasonably important and specific nuance to the data interpretation by Campbell et al.  The analysis is of a high quality, and the argument is valid; specifically, the LTE authors bring up the important distinction that between- and within-group changes (even when only one of two groups changes from pre to post) are not synonymous and shouldn't be interpreted the same way. The intent to treat perspective is also an interesting awareness for Sports Nutrition/Science researchers to be aware of from a cross-disciplinary perspective. This LTE is a positive outcome due to the original authors making their raw data accessible, and I commend the authors for doing so and the LTE authors for providing additional context and perspective on this study. 

Reviewer 2 Report

The Letter to the Editor concerning : Contrary to the Conclusions Stated in the Paper, Only Dry Fat-Free Mass was Different Between Groups upon Reanalysis of “Intermittent Energy Restriction Attenuates the Loss of Fat-Free Mass in Resistance Trained Individuals. A Randomized Controlled 7 Trial”

Peos and colleagues in this letter discussed two concerns:

- the authors have drawn conclusions from their data based on differences in nominal significance (i.e., the ‘DINS error’) rather than on actual differences between the diet groups.

- the authors have based their conclusion on people who completed the trial (a completers-only analysis, also known as a per protocol analysis), rather than also analyzing all people who were randomized to the diets (intention-to-treat analysis, also detailed below).

I suggest that this scientific discussion about data and analysis and the replication is of great importance and help scientists to understand and interpret the conclusion of the study and I recommend the publication of this letter.